

# Comparative analysis of BERT and FastText representations on crowdfunding campaign success prediction

Hakan Gunduz

Software Engineering Department, Kocaeli University, Kocaeli, Marmara, Turkey

## ABSTRACT

Crowdfunding has become a popular financing method, attracting investors, businesses, and entrepreneurs. However, many campaigns fail to secure funding, making it crucial to reduce participation risks using artificial intelligence (AI). This study investigates the effectiveness of advanced AI techniques in predicting the success of crowdfunding campaigns on Kickstarter by analyzing campaign blurbs. We compare the performance of two widely used text representation models, bidirectional encoder representations from transformers (BERT) and FastText, in conjunction with long-short term memory (LSTM) and gradient boosting machine (GBM) classifiers. Our analysis involves preprocessing campaign blurbs, extracting features using BERT and FastText, and evaluating the predictive performance of these features with LSTM and GBM models. All experimental results show that BERT representations significantly outperform FastText, with the highest accuracy of 0.745 achieved using a fine-tuned BERT model combined with LSTM. These findings highlight the importance of using deep contextual embeddings and the benefits of fine-tuning pre-trained models for domain-specific applications. The results are benchmarked against existing methods, demonstrating the superiority of our approach. This study provides valuable insights for improving predictive models in the crowdfunding domain, offering practical implications for campaign creators and investors.

## INTRODUCTION

The 2008 Global Economic Crisis prompted significant changes in economic behaviors, including the adoption of crowdfunding as an alternative funding model. Crowdfunding has become a vital method for start-ups to raise capital, especially during the early stages of development, when traditional funding sources such as angel investors and venture capital may be inaccessible (*Belleflamme, Omrani & Peitz, 2015*). This method allows entrepreneurs to reach potential investors *via* online platforms, combining small contributions to support various initiatives, from local park designs to the creation of video games (*Moritz & Block, 2016*). Kickstarter, a leading crowdfunding platform, has facilitated more than $7 billion in funding for hundreds of thousands of campaigns as of 2022 (*Polatos & Kernitskyi, 2023*).

Corresponding author
Hakan Gunduz,
hakan.gunduz@kocaeli.edu.tr

Despite its popularity, crowdfunding faces challenges such as information asymmetry, financing uncertainty, and low success rates (*Moradi et al., 2023*). Campaign blurbs play a crucial role in mitigating information asymmetry by providing textual signals that influence financing performance. Although personal social networks and geography can also reduce information asymmetry, many investors struggle to identify campaign quality, leading to risks for entrepreneurs and a market where low-quality campaigns might overshadow high-quality ones. Sentiment analysis using artificial intelligence (AI) models can offer efficient solutions to predict crowdfunding performance by uncovering patterns in backer sentiments. This approach not only helps platforms assess campaign sentiments, but also helps backers make informed investment decisions, enhancing their participation in ventures that align with positive sentiments (*Ryoba et al., 2020*, *Hu & Yang, 2020*).

This study addresses the research gap for the use of advanced AI models, specifically bidirectional encoder representations from transformers (BERT) and FastText, for sentiment analysis in crowdfunding. By converting campaign blurbs into feature vectors with these models and applying long short term memory (LSTM) and gradient boosting machine (GBM) for classification, we aim to investigate the performance of different BERT layers in improving the accuracy of the prediction, providing a comprehensive analysis of their impact. In this regard, our study introduces two prominent features that distinguish it from previous NLP-based crowdfunding prediction studies:

First, we extract various qualitative representations from the 12-layer BERT architecture, allowing us to evaluate the performance of these various layer feature representations. Furthermore, sentiments can vary across campaign stages and evolve over time, necessitating the use of dynamic analysis techniques. This method allows us to gain insight into the effect of different layers in BERT on the classification task.

Second, we investigate the use of various types of machine learning models for classification, pairing them with various representations. We identify the best attribute representation and synthetic model pair through experimentation with various combinations of representations and models. We aim to leverage the strengths of deep contextual embeddings and robust machine learning models. This research not only demonstrates the superior performance of BERT in sentiment analysis and prediction tasks, but also provides insights into the practical applications of these models in the crowdfunding domain.

The organization of the article is as follows. "Related Works" reviews related work in the field of crowdfunding and sentiment analysis. "Materials and Methods" details the materials and methods used, including the dataset, preprocessing steps, and the AI models employed. "Experimental Results" presents the experimental results. Finally, "Conclusion and Discussion" concludes the article and suggests directions for future research.

## RELATED WORKS

Crowdfunding has experienced significant growth, with platforms like Kickstarter becoming central to reward-based crowdfunding. This surge has drawn considerable attention from researchers, particularly within computer science. Researchers have traditionally relied on meta-features from campaign profiles to build AI models for

predicting campaign outcomes. For instance, *Greenberg et al. (2013)* used features such as sentence count, campaign duration, and creator demographics, training support vector machines (SVM), and decision trees to classify campaign statuses. Although these models are effective in certain contexts, they often fail to capture the nuanced language and sentiment expressed in campaign blurbs.

More advanced approaches have focused on extracting data directly from campaign web pages and social media. *Etter, Grossglauser & Thiran (2013)* utilized time series and Twitter data to predict the success of the Kickstarter campaign, demonstrating the importance of real-time data and social influence. *Buttice & Rovelli (2020)* explored the impact of narcissism on crowdfunding success, highlighting the role of psychological factors. Various models, including deep learning, decision trees, logistic regression, and K-nearest neighbors (KNN), have been used to predict funding success, achieving high accuracy rates (*Gülşen et al., 2015*; *Yu et al., 2018*; *Wang, Zheng & Wu, 2020*; *Hongke et al., 2017*) analyzed social media posts to understand campaign dynamics, emphasizing the importance of external validation and engagement metrics.

Sentiment analysis, which involves the use of natural language processing (NLP) and machine learning to classify text sentiment, has gained prominence in crowdfunding research. This technique helps uncover the overall sentiment of backers and potential backers, providing insights that campaigners can use to craft more compelling pitches. *Hu & Yang (2020)* and *Jhaveri et al. (2019)* emphasized the importance of analyzing textual descriptions to predict campaign outcomes, demonstrating that sentiment can be a powerful predictor of success.

Advanced textual analysis of campaign blurbs and social media posts has also been explored. *Westerlund et al. (2019)* used topic modeling on a dataset of 21,188 technology campaigns from Kickstarter to predict fundraising success, finding that campaign summaries alone could indicate success likelihood. *Faralli et al. (2021)* developed a Need Index-based model to predict the success of the Kickstarter campaign, achieving a prediction accuracy of 94.4%. *Wang, Guo & Wu (2022)* analyzed the predictive power of internet reviews on fundraising outcomes, finding that deep learning methods were more effective than shallow learning techniques.

The use of BERT as a classification embedding tool has gained considerable attention in recent years. BERT is a deep learning model designed to understand the context of a word in search queries, which has shown exceptional performance in various NLP tasks (*Devlin et al., 2018*). Unlike traditional models, BERT considers the context of a word by looking at the words that come before and after it, making it particularly powerful for tasks involving language understanding. Numerous studies have leveraged BERT for tasks such as sentiment analysis, question answering, and text classification, demonstrating its ability to outperform earlier models such as Word2Vec and GloVe (*Saleh, Alhothali & Moria, 2023*; *Rakshit & Sarkar, 2024*). However, its application in predicting crowdfunding success through campaign blurbs remains underexplored.

FastText, another word embedding model, represents words as n-grams and has been effective in handling out-of-vocabulary words and morphologically rich languages (*Joulin et al., 2016*). It has been widely used in text classification tasks due to its efficiency and the

quality of its embeddings. Although FastText has shown good performance in various applications (*Sadiq, Aljrees & Ullah, 2023*; *Umer et al., 2023*; *Raza et al., 2023*), the comparative effectiveness of FastText and BERT in the context of crowdfunding prediction needs further investigation.

# MATERIALS AND METHODS

This section presents the approach and techniques used to analyze crowdfunding data and investigate the relationship between sentiment analysis and fundraising outcomes. First, we describe the dataset used in this research, including the specific crowdfunding platform and the campaign data collected. Next, we outline the sentiment analysis process, which involves the application of natural language processing and machine learning techniques to extract subjective information from campaign blurb data.

## Dataset

The dataset used in this study was sourced from a public GitHub repository (https://github.com/rajatj9/Kickstarter-projects) and comprises approximately 191,724 Kickstarter campaigns posted in 2018. The dataset includes various campaign characteristics such as the campaign name, blurb, funding date, funding period, amount of funding, and campaign success status. For this research, the campaign blurb is used as the input variable, and the campaign success status is the target variable for our machine learning models. Figure 1 shows the class distribution of the prediction variable, revealing that successful campaigns are fewer than failed ones by around 40,000.

Since campaign blurbs are used to predict success status, we visualized the relationships between words in the blurbs and success status. Word clouds for successful and failed campaigns are shown in Figs. 2 and 3. Dominant words in successful campaigns include 'world', 'help' and 'one', while failed campaigns often contain 'new', 'will' and 'project'.

To use campaign definitions in machine learning models, they must be vectorized. We created a word histogram for the campaign blurbs, shown in Fig. 4, and set a 25-word limit for each blurb based on the mode value of the histogram. Texts longer than 25 words were trimmed and shorter texts were padded with a unique token to ensure consistent length.

## Text processing techniques

### FastText

FastText is a technique to categorize and convert text into vectors for use in natural language processing (NLP) tasks. Words are represented as character n-grams in this approach, which then use a shallow neural network structure to get compact vector representations of the words in a continuous space. This approach is useful in a range of applications due to its computational efficiency, ability to handle out-of-vocabulary (OOV) words, pre-trained embedding support, and native text classification properties.

FastText is a useful tool for natural language processing since it can encode words using character n-grams, allowing morphological and semantic features to be included even for words that are not often seen in texts. FastText also supports pre-trained embeddings, which might be beneficial in some cases of text analysis. Furthermore, the application has a

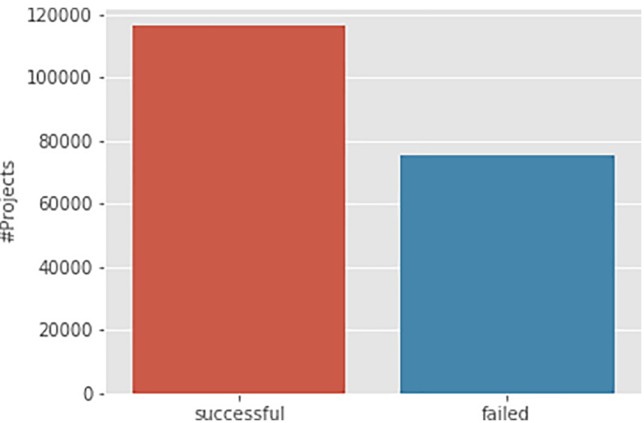

**Figure 1 Class distribution of Kickstarter campaigns.**

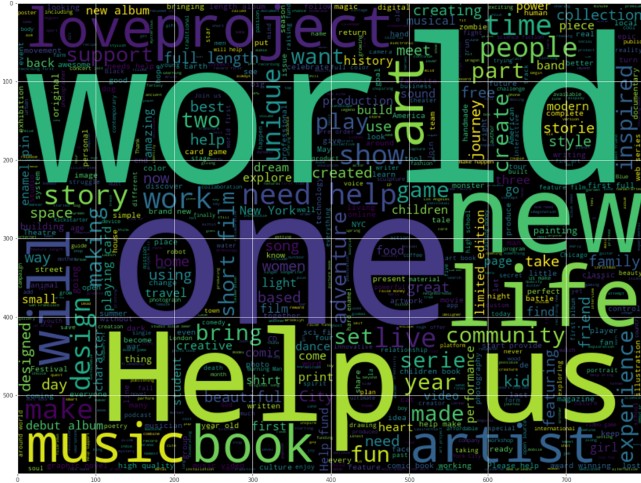

**Figure 2 Word cloud of successful campaigns.**

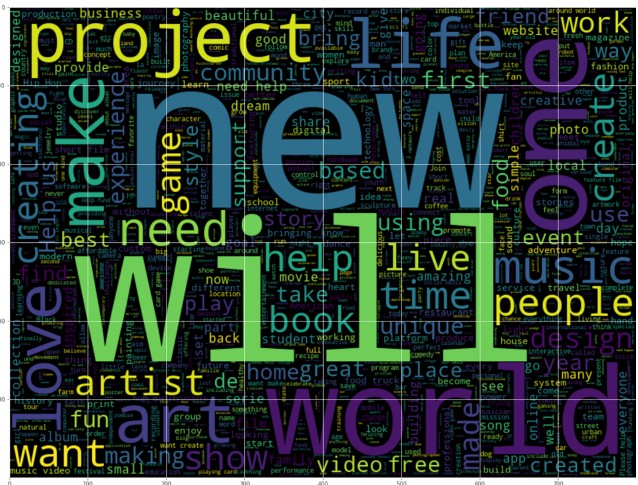

**Figure 3 Word cloud of failed campaigns.**

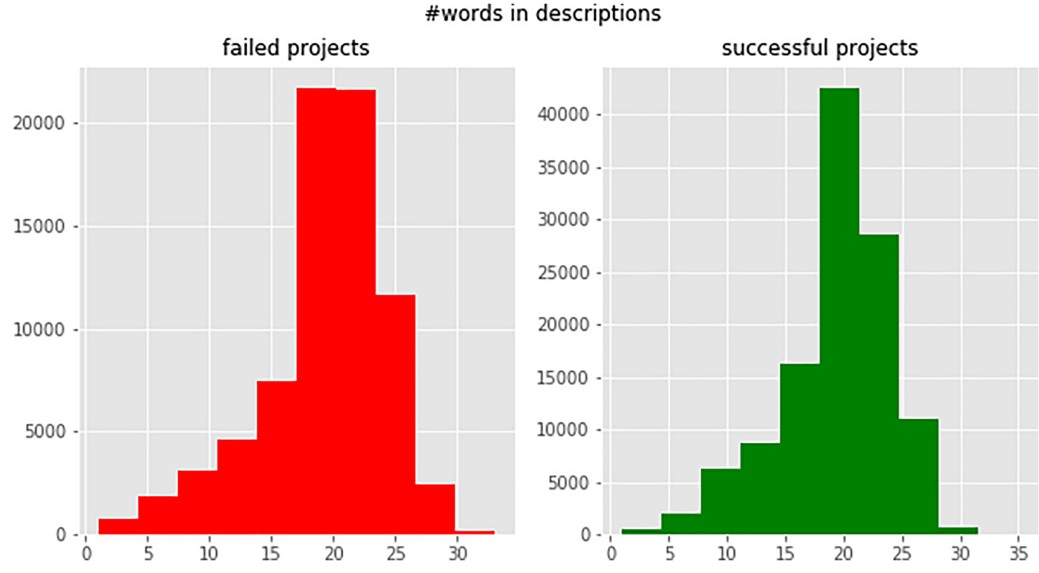

**Figure 4 Number of words per campaign blurb (successful and failed campaigns).**

classification capability, which makes it ideal for a variety of research campaigns that demand rapid and effective text analysis (*Joulin et al., 2016*).

### Bidirectional encoder representations from transformers

Bidirectional Encoder Representations from Transformers, a language model based on transformer architecture, is a revolutionary approach to language modeling. Unlike traditional models that predict the next word based on preceding words, BERT uses both the previous and the next words to predict. Although the left-to-right approach is effective in predicting individual words, BERT distinguishes itself by masking random words and attempting to guess them. This innovative methodology promises to transform the field of text generation (*Devlin et al., 2018*).

Immediately preceding the BERT model, there were language models, which utilized a large unlabeled *corpus*. However, they were unable to handle compound words and negations and did not consider the context in which the words were used. BERT, on the other hand, employs deep learning architectures, specifically transformer architecture, to overcome these shortcomings. Attention is applied to the words in the context, thus creating word representations that take into account the other words occurring. BERT comprises a multiple of layers that include both encoder and decoder layers. However, the initial BERT model uses only the encoder layers for pre-training and fine-tuning (*Tenney, Das & Pavlick, 2019*).

BERT's encoder layer comprises different types of embeddings. The initial step in the process of token embeddings involves breaking down the input text into discrete units known as tokens, either individual words or sub-words. Each token is then linked to a specific token embedding that captures its semantic meaning. Incorporating segment embeddings, BERT enables input with multiple sentences or segments, and each token is assigned a distinct segment embedding to signify its owing to a particular segment. The

incorporation of position embeddings is necessary in BERT as it does not process text in a linear fashion, and thus requires the capturing of the positional information of tokens. Each token is assigned a position embedding to encode its place in the input sequence.

The internal structure of the BERT architecture is shown in Fig. 5. Transformer encoder layers utilized by BERT, comprise a sequence of several transformer encoder layers, each of which is composed of two distinct sub-layers: a multi-head self-attention mechanism and a position-based, fully connected feed-forward network. The sub-layer known as Multi-Head Self-Attention facilitates tokens in attending to other tokens within the input sequence, capturing the intrinsic contextual relationships between them. Through this process, it computes the attention weights for each token based on its connections to other tokens. Following the implementation of self-attention, a position-wise feed-forward network is utilized to process each token individually. This network involves two linear transformations separated by a ReLU activation function. The ultimate tier of BERT's encoder is responsible for producing the final representations. During pre-training, BERT anticipates the masked tokens by utilizing a softmax classification layer across the token embeddings. However, during fine-tuning, additional task-oriented layers are incorporated on top of the BERT encoder to accomplish precise NLP tasks such as sentiment classification or named entity recognition. The number of encoder layers in BERT varies depending on the specific model variant employed, ultimately impacting the model's ability to capture intricate linguistic patterns and contextual cues.

### Preprocessing steps

The preprocessing steps are essential for transforming raw textual data from campaign blurbs into structured input suitable for machine learning models. These steps ensure that the data are clean, consistent and in a format that can be used effectively to train and evaluate predictive models.

1. **Data collection:** We collected the dataset from a public GitHub repository, which included 191,724 Kickstarter campaigns posted in 2018.
2. **Text cleaning:** We preprocessed campaign blurbs by removing punctuation and stop words and performing tokenization to break down the text into individual words.
3. **Word limits:** Each campaign blurb was limited to 25 words, trimming longer texts and padding shorter ones to maintain consistency.
4. **Feature extraction:**

   - **FastText:** Pre-trained FastText representations trained on Wikipedia articles were used. The average of these vectors transformed campaign blurbs into 300-dimensional feature vectors. FastText was chosen for its computational efficiency and ability to handle out-of-vocabulary words using character n-grams, making it valuable for diverse and unique campaign blurbs (*Alomari & Ahmad, 2024*).
   - **BERT:** The pre-trained BERT model from Huggingface was used. Campaign blurbs were tokenized using the Bert Tokenizer, generating 768-dimensional feature representations. Features were extracted from the first, last, and all layers of BERT,

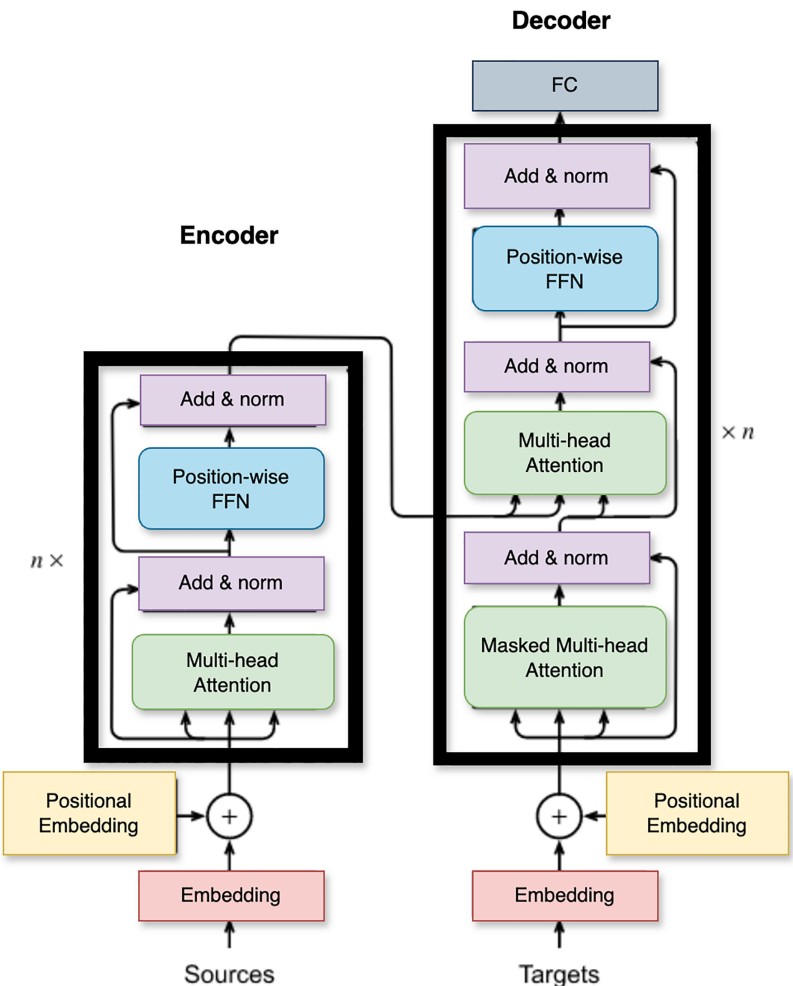

**Figure 5** **The encoder-decoder structure of the BERT architecture.**

with representations from multiple layers averaged when more than one layer was used. We selected BERT for its ability to capture deep contextual relationships in text through its multi-layer transformer architecture. The choice of specific BERT layers (first, last, and all) was based on their ability to capture different levels of linguistic features, from shallow syntactic structures to deeper semantic meanings. Fine-tuning BERT with our dataset allowed better adaptation to the specific language patterns in crowdfunding blurbs (*Hao et al., 2019*).

The sequence of words fed into the LSTM is critical as the LSTM models capture the temporal dependencies and order of words in a sentence. In our approach, each campaign blurb was tokenized and the first 25 words were selected to maintain a consistent input length for the models. This decision was based on the following considerations:

1. **Consistency and computational efficiency:** Limiting the input length to 25 words ensures that all input vectors are of the same dimension (25*768 for BERT). This

uniformity simplifies model training and reduces computational complexity, making the process more efficient.

2. **Focus on key information:** Kickstarter campaign blurbs typically highlight the most crucial information within the first few sentences. By focusing on the first 25 words, we capture the essence of the campaign without including redundant or less relevant details that may appear later in the text.

3. **Handling longer texts:** For campaign blurbs longer than 25 words, we trim the text to the first 25 words. This approach ensures that the model processes a manageable and consistent input size, reducing the risk of overfitting and improving generalizability.

4. **Preservation of contextual flow:** The order of the words is preserved as they appear in the original blurb. This method allows the LSTM to capture the sequential dependencies and contextual flow within the truncated text, which is crucial for understanding sentiment and predicting campaign success.

5. **Padding shorter texts:** For campaign blurbs shorter than 25 words, we use padding to ensure that all input vectors are of the same length. This padding does not affect the sequential information captured by the LSTM, as the padding token is designed to be neutral.

Our methodology differs from some past research papers that may use the entire text input directly into BERT with LSTM. However, our approach balances computational efficiency and model performance, ensuring that the most relevant information is captured while maintaining manageable input sizes for the models.

## Machine/deep learning algorithms

### Gradient boosting machines

Gradient boosting machines are advanced machine learning models that combine multiple less powerful predictive models (typically decision trees) to create a robust and accurate prediction model. GBM uses gradient descent optimization to minimize residuals, adapting and handling complex or noisy datasets effectively (*Friedman, 2001*).

GBMs includes a variety of hyperparameters that may be modified to improve model performance, such as learning rate, tree depth, and the number of trees in the ensemble. The gradient descent optimization process can be controlled by adjusting the learning rate, but the depth of the tree and the number of trees could affect the complexity of the model and the ability to spot patterns in the data (*Natekin & Knoll, 2013*).

GBM was chosen for its robustness and ability to handle a wide range of data formats, making it suitable for complex predictive modeling tasks.

### Long-short term memory

Long-short term memory, a type of recurrent neural network (RNN), is designed to capture time series relationships. It includes mechanisms for adding or removing information from the cell state, using gates to control the flow of information. This structure allows LSTM to handle long-term dependencies and sequential data effectively, making it suitable for text classification tasks.

LSTM generates an output vector ($h_t$) for each time step in the time series data to connect the output vector of the current time step to the previous time steps. The most common use of LSTM is to represent an entire series using the output vector of the final time step of the sequence ($h_t$). Because the entire sequence is converted to a low-dimensional vector, this procedure may result in information loss. In certain cases, instead of taking the vector from the most recent time step, you can utilize the output vectors of all previous phases (*Gunduz, 2021*).

LSTM was selected because of its effectiveness in capturing sequential dependencies in text, which is crucial for understanding the context and sentiment expressed over multiple words. The sequence of the 25 words fed into LSTM was maintained in the order in which they appeared in the campaign blurbs, ensuring the preservation of contextual flow.

## Evaluation metrics

Although accuracy is a recommended measure in performance evaluation, it does not examine class discrimination. F-Measure is an alternative statistic to validate the performance of a class-based model assessment. The accuracy and F-measure computations are closely related to the confusion matrix (CM), which basically provides the number of accurate and wrong predicted occurrences per class (Table 1). The values utilized to compute the aforementioned metrics are true positive (*tp*), false positive (*fp*), false negative (*fn*), and true negative (*tn*) (*Gunduz, 2022*).

Accuracy is described as the proportion of accurate estimations to the total number of instances. However, when the ratio of *fp* to *fn* increases very much, the F measure must be used in the performance evaluation.

F-measure takes the harmonic mean of precision and recall. As a result, false positive and false negative samples are used to determine discrimination between classes. The F-measure is calculated using the confusion matrix as in Eqs. (1)–(3):

$$\text{precision} = \frac{tp}{tp + fp} \tag{1}$$

$$\text{recall} = \frac{tp}{tp + fn} \tag{2}$$

$$\text{F-Measure} = \frac{2 \times precision \times recall}{precision + recall}. \tag{3}$$

## EXPERIMENTAL RESULTS

Figure 6 presents a graphical abstract of our proposed model, which consists of four steps. First, the campaign definitions and funding status were acquired from the dataset file. This step included the removal of punctuation and the omitting of words not defined in the dictionary from the campaign definition text. In the last process, the campaign texts were divided into tokens and an ID was assigned for each word. In the second step, word representations of each token in the campaign definitions were produced using pre-trained FastText and BERT models. The third step involved converting the word representations into a format suitable for input into machine learning models. In the fourth step, the input

| Table 1 Confusion matrix for two-class classification. | | |
|---|---|---|
| **Actual/Predicted as** | **Positive** | **Negative** |
| Positive | *tp* | *fn* |
| Negative | *fp* | *tn* |

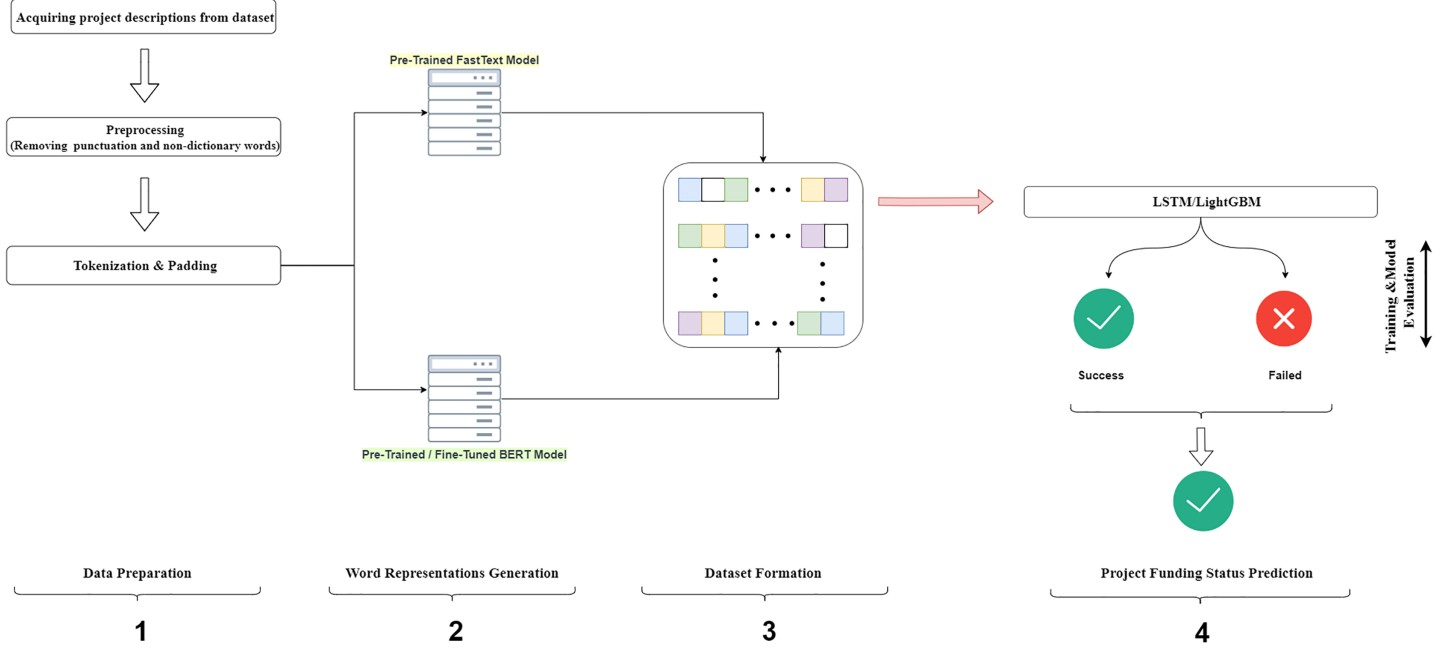

**Figure 6 Diagram of the proposed model.**

data was fed into the LSTM and GBM models for training. This step also included the optimization of the model parameters. After training the models with the best set of parameters using the training data, the performance of the models was evaluated with the test set using evaluation metrics. The hold-out method was used to evaluate the performance of the models, with 80% of the dataset randomly assigned as the training set and 20% as the test set, maintaining class distributions.

To ensure the reproducibility of our experiments, we provide detailed specifications for the software and hardware used in our study. The experiments were carried out using Python version 3.11.3, and the deep learning models were implemented with TensorFlow version 2.4.1 and the Huggingface Transformers library version 4.5.1. The computations were performed on a system equipped with an NVIDIA GeForce GTX 1070 GPU with 8 GB of memory, running on Ubuntu 20.04 LTS as an operating system. The CPU used was an Intel Core i7-7700K. Furthermore, the system had 16 GB of RAM, which facilitated the handling of the dataset and complex model training processes. All necessary Python packages and libraries were managed using the Anaconda distribution.

**Table 2  Parameters of LSTM network.**

| Parameters | Value |
| --- | --- |
| Number of memory cells | {64, 128, 256} |
| Dropout rate | {0.2, 0.3, 0.4} |
| Recurrent dropout rate | {0.1, 0.2} |
| Optimizer | Adam |
| Activation function | {RELU, ELU} |
| Number of epochs | 50 |
| Batch size | 32 |

**Table 3  Parameter space of LightGBM.**

| Parameters | Value |
| --- | --- |
| Number of learners | {100, 200, 300, 400, 500, 1,000} |
| Learning rate | {0.1, 0.01} |
| L2-regularizer | {0.001, 0.0001} |
| Max_depth | {7, 9, 11} |

## FastText representations

In the initial trials, FastText representations were used to convert campaign blurbs into feature vectors. Pre-trained FastText representations trained on Wikipedia articles were applied throughout the conversion process. The FastText vector representations of each word were obtained and the averages of these vectors were taken to transform the campaign blurbs, limited to 25 words, into feature vectors. Consequently, each campaign document was transformed into a 300-dimensional feature vector. The models employed have several parameter settings that must be adjusted. To find the optimal parameter set, 20% of the training set was designated as the hold-out set for grid search. Tables 2 and 3 list the parameters and their value ranges used in this process.

FastText feature vectors were fed into two different machine learning algorithms for classification. While the GBM model received 300-dimensional mean vectors, the LSTM model took vectors in a 2D format of 25*300. Note that 25 denotes the number of words in the blurbs and 300 denotes the size of the FastText vector for each word. Table 4 summarizes the classification results achieved using FastText representations.

The results obtained with FastText representations showed that the LSTM model had higher classification success than GBM. The LSTM model achieved an F-Measure of 0.772 with an accuracy of 0.714, while the GBM model's accuracy was 0.692.

## BERT representations

In the second experiment, the BERT model was applied in two different ways. First, we used the pre-trained BERT model from Huggingface as a feature extractor. Second, we fine-tuned BERT with our dataset. To prevent the fine-tuning process from dominating the

**Table 4 Classification results with FastText representations.** The bold entries in highlight the highest performance achieved using FastText representations.

| | GBM | | LSTM | |
|---|---|---|---|---|
| | Acc. | F-Mea. | Acc. | F-Mea. |
| FastText Repr | 0.692 | 0.772 | **0.714** | **0.772** |

**Table 5 Classification results with non-tuned and fine-tuned BERT.** The bold entries represent the best performance results obtained using BERT representations.

| | GBM | | LSTM | |
|---|---|---|---|---|
| | Acc. | F-Mea. | Acc. | F-Mea. |
| BERT Pre-trained (Non-tuned) | 0.707 | 0.779 | 0.711 | 0.781 |
| BERT Fine-tuned (First layer) | 0.712 | 0.774 | 0.720 | 0.780 |
| BERT Fine-tuned (Last layer) | 0.732 | 0.793 | 0.737 | 0.800 |
| BERT Fine-tuned (All layers mean) | **0.738** | **0.800** | **0.745** | **0.804** |

network weights, the learning rate during training was set at 1e-5. BERT comprises 12 transformer layers, with the first layer known as the embedding layer. During the tests with fine-tuned BERT, the representations derived from various levels of the BERT model were fed into the models. The chosen layers included the first, last and all 12 layers. Campaign blurbs were converted to the input format of the BERT model using the Bert Tokenizer, generating 768-dimensional feature representations for each blurb at each layer. When multiple layers were used in an experiment, the representations were averaged. While the GBM model received an average of 25-word vectors, the LSTM model received 2D input with 25*768 dimensions, as in the FastText experiments. Table 5 presents the classification results achieved using pre-trained and fine-tuned BERT representations.

The highest classification success was achieved with the combination of the LSTM model and the mean of all layer representations. This model reached an accuracy of 0.745 and an F-Measure of 0.804. The GBM model that used all layer representations achieved an accuracy of 0.738 and an F-Measure of 0.800. The features extracted from the last layer of BERT showed lower performance compared to the features from all layers. Training the LSTM model with the last layer features resulted in an accuracy of 0.732 and an F-measure of 0.793. The features of the first layer performed lower than the last layer and all layers, with the LSTM and GBM models achieving accuracies of 0.712 and 0.720, respectively.

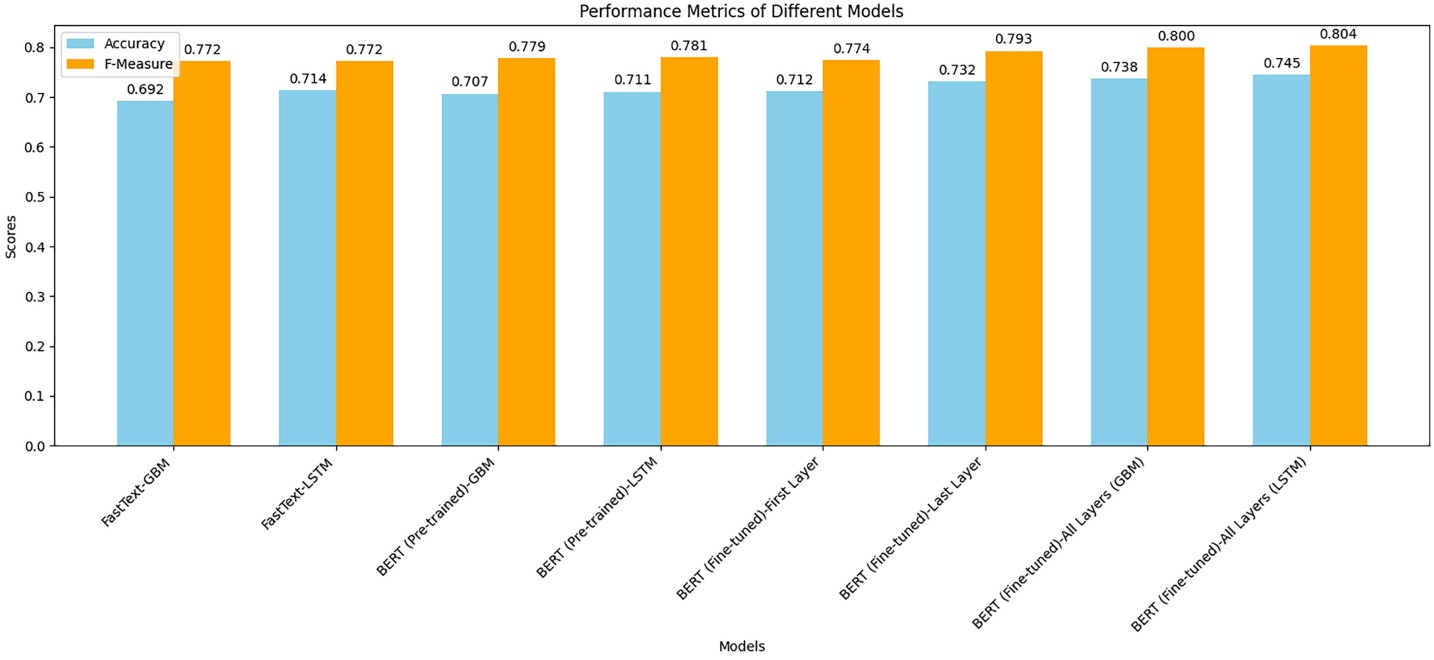

**Figure 7  Comparison of the performance metrics for all models.**

Classification performance with non-tuned BERT was lower than with fine-tuned BERT. The LSTM model with non-tuned BERT representations achieved an accuracy of 0.711 and an F-measure of 0.781.

Figure 7 shows a combined bar chart comparing the performance metrics (accuracy and F-measure) for different models. This chart provides a clear visual comparison of how each model performed across the two metrics. Notably, the fine-tuned BERT models, especially when combined with the LSTM classifier, achieved the highest scores in both accuracy and F-measure. The chart illustrates the significant performance improvement achieved by fine-tuning BERT compared to using pre-trained BERT or FastText representations. The LSTM models generally outperformed the GBM models, indicating their effectiveness in capturing sequential dependencies in text data. This visual representation underscores the superior performance of BERT representations, particularly when fine-tuned, in predicting crowdfunding campaign success.

Figure 8 presents a comprehensive comparison of performance metrics between the best-performing models: LSTM with BERT representations (all layers) and GBM with BERT representations (all layers). This graph illustrates several key metrics, including accuracy, precision, recall, and F1-score for both classes (Class 0 and Class 1).

The LSTM model with BERT representations (all layers) achieved the highest overall accuracy of 0.745. For Class 0 (failed campaigns), the precision was 0.721, recall was 0.569, and F1-score was 0.636. For Class 1 (successful campaigns), the precision was 0.755, recall was 0.859, and F1-score was 0.803. These metrics indicate that the LSTM model is

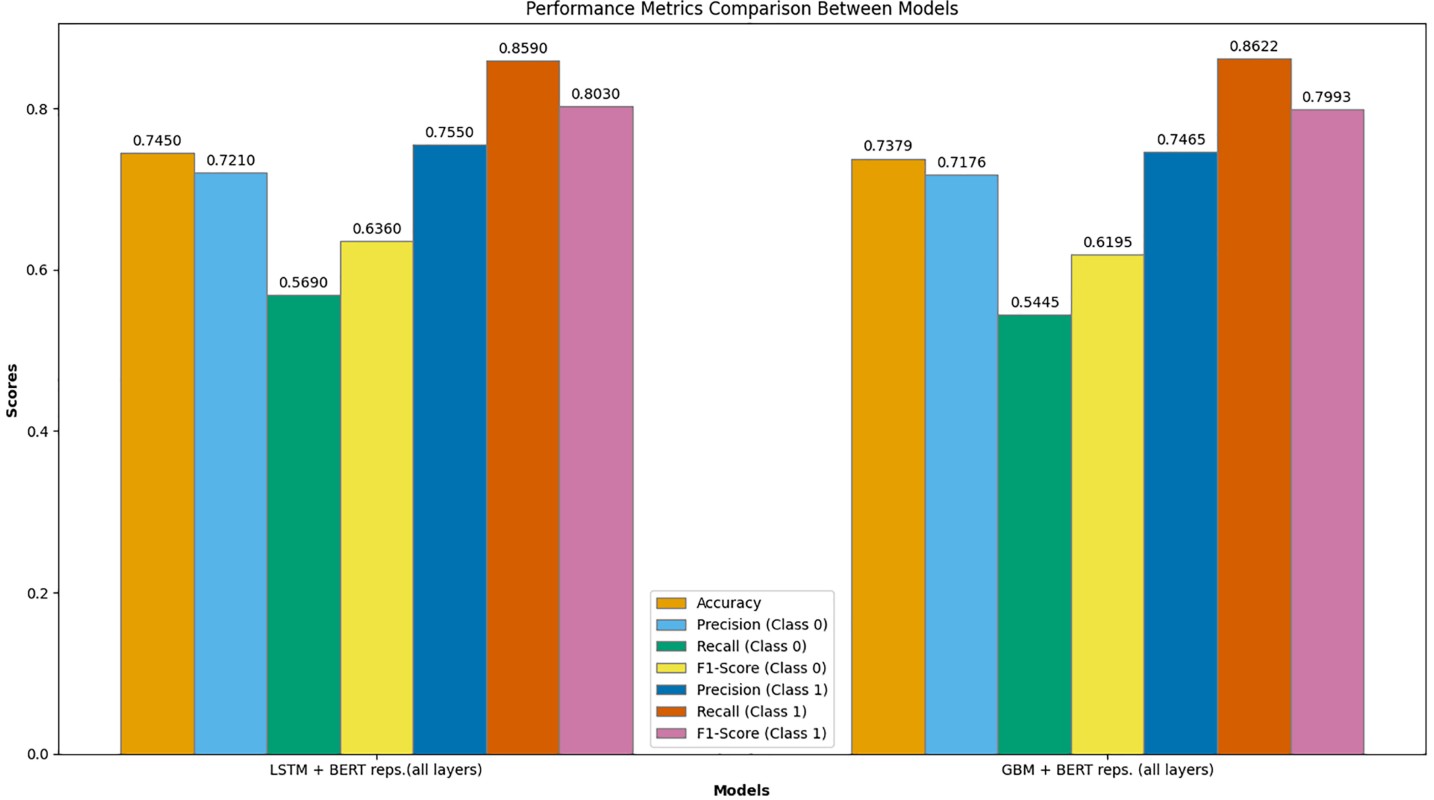

**Figure 8 Comparison of the performance metrics between LSTM with BERT representations (all layers) and GBM with BERT representations (all layers).**

**Table 6 Recent studies that used text to predict crowdfunding success.** The bold entries indicate the performance metrics of the current study, which are directly comparable to those of previous studies listed.

| Study | Accuracy | F-Measure |
|---|---|---|
| *Lee, Lee & Kim (2018)* | 0.765 | NA |
| *Kaminski & Hopp (2020)* | 0.710 | 0.710 |
| *Tang et al. (2022)* | 0.722 | NA |
| **Our study** | **0.745** | **0.801** |

particularly effective at predicting successful campaigns, as evidenced by the high recall and F1-score for Class 1.

In comparison, the GBM model with BERT representations (all layers) achieved an accuracy of 0.738. For Class 0, the precision was 0.718, recall was 0.5445, and F1-score was 0.620. For Class 1, the precision was 0.747, recall was 0.862, and F1-score was 0.799. While the GBM model also performs well, particularly for successful campaigns, it slightly lags behind the LSTM model in overall accuracy and performance metrics for Class 0.

This detailed comparison highlights the superior performance of the LSTM model with fine-tuned BERT representations across multiple evaluation metrics, demonstrating its robustness and effectiveness in predicting crowdfunding campaign success.

# CONCLUSION AND DISCUSSION

This section contains the following subsections: First, we summarize the key findings of our study, highlighting the performance differences between the models and the impact of fine-tuning. Next, we discuss the limitations encountered during our research, focusing on dataset constraints and potential biases. We then explore the implications of our findings for the crowdfunding industry, detailing how our results can inform better decision-making for both investors and campaign creators. Following this, we integrate our findings with recent studies to contextualize our contributions within the broader field. Finally, we outline future research directions that can build upon our work, suggesting ways to enhance model performance and address current limitations.

## Key findings

In this study, we explored the effectiveness of advanced AI techniques in predicting the success of Kickstarter crowdfunding campaigns by analyzing the sentiments conveyed in campaign blurbs. Our comparative analysis of BERT and FastText representations, combined with the LSTM and GBM models, revealed that BERT representations significantly outperform FastText. The highest accuracy rate of 0.745 was achieved by combining all BERT layer representations with an LSTM model. Fine-tuning the pre-trained BERT model with our dataset resulted in a 3% performance improvement, underscoring the importance of model customization to the specific dataset.

- **Performance of BERT *vs*. FastText:** BERT representations consistently outperformed FastText across all metrics, demonstrating the advantage of deep contextual embeddings.
- **Fine-tuning benefits:** Fine-tuning the BERT model with our specific dataset improved performance, highlighting the value of adapting pre-trained models to domain-specific data.
- **Model comparisons:** LSTM models, which handle sequential data effectively, showed slightly better performance than GBM models, particularly when combined with BERT representations.

## Limitations

- **Dataset size:** Our study was conducted using a dataset of 191,724 Kickstarter campaigns from 2018, which, while substantial, may not capture all the nuances and trends present in more recent data or other crowdfunding platforms.
- **Potential biases:** The analysis is based solely on textual data from campaign blurbs, which may introduce biases related to language usage and presentation style. Visual and social media data were not considered in this study.

- **Generalizability:** The findings may not generalize to other crowdfunding platforms with different user bases and campaign characteristics.

## Implications for crowdfunding industry

Our findings have several implications for the crowdfunding industry. By leveraging advanced AI techniques, platforms can improve their ability to predict campaign success, helping investors make more informed decisions and potentially reducing market risk. Campaign creators can also benefit from insights into how the language and sentiment of their blurbs affect backer engagement. Moreover, this study emphasizes the importance of linguistic and emotional factors in the presentation of crowdfunding campaigns, suggesting that creators should carefully craft their blurbs to enhance appeal and potential success.

## Integration with recent studies

Table 6 lists recent studies that used text to predict the success of crowdfunding.

In these studies, texts were transformed into feature vectors with GloVe representations and BERT and given as input to deep learning models. Campaign blurbs were converted to feature vectors with GloVe representations and classified with two different deep learning architectures in *Lee, Lee & Kim (2018)*, achieving a classification success of 0.765 with the SeqtoSeq model. In another study, the success of campaigns was classified by extracting features from different types of data (video and text) on crowdfunding profile pages (*Kaminski & Hopp, 2020*; *Tang et al., 2022*). Although features in texts were produced with GloVe representations, the ResNet architecture was used to extract features from video data. These two models were fed into the Deep Cross-Attention Network model, and the classification status of the campaigns was predicted. The highest success rate in the Kickstarter data for this study was 0.722.

Although our study has close accuracy rates with the aforementioned studies, what distinguishes it from these studies is the use of different embedding methods in the classification process and the assessment of these methods with different ML models. In this context, the LSTM and GBM models were trained with BERT and FastText representations, and the performances of the models were comparatively analyzed. While LSTM successfully models sequence data, GBM stands out for its resistance to overfitting. Additionally, a fine-tuning process was applied to the pre-trained BERT model, and feature representations were extracted from different layers of this model. In this respect, it is the first study to evaluate the performance of BERT-extracted layer representations in crowdfunding success prediction studies.

## Future research directions

- **Incorporating multimodal data:** Future research should explore integrating textual, visual, and social media data to provide a more comprehensive analysis of crowdfunding campaign success. This approach could offer a more holistic view of how different types of content influence backer decisions.

- **Longitudinal studies:** Conducting longitudinal studies to analyze how campaign success predictors evolve over time and across different economic conditions. Such studies could provide valuable insights into the changing dynamics of crowdfunding and the long-term effectiveness of different predictive models.
- **Scalability:** Exploring the scalability of these AI models across different crowdfunding platforms to validate their generalizability and effectiveness. This would involve testing the models on various platforms with diverse user bases and campaign types to ensure robust performance.
- **Comparative analysis with other models:** Further research could include comparative analysis with other advanced models such as Transformer-based models beyond BERT, including GPT and its variants, to assess their efficacy in predicting crowdfunding success.

## Discussion

Our results highlight the potential of AI-driven sentiment analysis in enhancing the prediction of crowdfunding campaign success. The superior performance of BERT representations, particularly when fine-tuned, demonstrates the value of deep contextual embeddings in capturing the nuances of campaign language. The combination of BERT with LSTM models, which are adept at handling sequential data, proved particularly effective, achieving the highest accuracy.

The study also underscores the importance of fine-tuning pre-trained models to specific datasets. The 3% improvement in performance with fine-tuned BERT models indicates that domain-specific customization can significantly enhance predictive accuracy. This finding is crucial for the application of AI in real-world scenarios, where generic models may fall short without adequate adaptation.

However, the limitations of the study, including its reliance on textual data and the potential biases inherent in language usage, suggest areas for improvement. Future research integrating multimodal data could provide a more comprehensive understanding of campaign success factors. In addition, addressing biases in AI models is essential to ensure fair and equitable outcomes.

Our study contributes to the growing body of research on AI applications in crowdfunding by demonstrating the superior performance of BERT representations and the benefits of fine-tuning models. These findings pave the way for more advanced and comprehensive approaches to predict crowdfunding success, ultimately benefiting both investors and campaign creators.

### Funding

The authors received no funding for this work.

### Competing Interests

The authors declare that they have no competing interests.

## Author Contributions

- Hakan Gunduz conceived and designed the experiments, performed the experiments, analyzed the data, performed the computation work, prepared figures and/or tables, authored or reviewed drafts of the article, and approved the final draft.

## Data Availability

The code are available at GitHub and Zenodo:

- https://github.com/hakangunduz86/Crowdfunding-Success-Prediction
- Gunduz, H. (2023). Kickstarter 2018 (Pre-Processed Data) [Data set]. Zenodo. https://doi.org/10.5281/zenodo.12735215.

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
