# Peer review of "Comparative analysis of BERT and FastText representations on crowdfunding campaign success prediction"

_PeerJ Computer Science, doi:10.7717/peerj-cs.2316_

## Round 0.1 · original submission · Major Revisions

Dear authors,

Thank you for your submission. Based on the reviewers’ comments, you may resubmit the revised manuscript for further consideration. Please consider the reviewers’ comments carefully and submit a list of responses to the comments along with the revised manuscript. Furthermore, it will be better to address the following in the revised paper:

1. Abstract does not clearly explain the contribution. The motivation of the paper does not exist. The contribution is not properly explained in an understandable way. The abstract section should be rewritten in order to clearly state the manuscript's main focus. The abstract should give the readers essential details, i.e., including the main contributions, the proposed method, the main problem, the obtained results, the benchmark tests, the comparative methods, etc. Efforts are needed to make the abstract coherent while clearly describing the problem being investigated and findings.
2. The current introduction is simple and misses many contents related to the problem formulation. Please write research gap and the motivation of the study. Evaluate how your study is different from others.
3. In general, the literature review is not sufficient. It is more of the type “researcher X did Y” rather than an authoritative synthesis assessing the current state-of-the-art. Where do we stand today? What approaches are there in the literature to model the problem? What are the main differences between them? What are their weaknesses and strengths? Why they are better or how? What’s new/novel here?
4. Organization of the paper should be given at the end of Introduction section.
5. Equations should be cited with equation number, not as “below”, “as follows”, and etc. This should also be addressed for figures and tables in the text.
6. The paper lacks the running environment, including software and hardware. The analysis and configurations of experiments should be presented in detail for reproducibility. It is convenient for other researchers to redo your experiments and this makes your work easy acceptance. A table with parameter settings for experimental results and analysis should be included in order to clearly describe them.

·

Basic reporting

Review of "Comparative Analysis of BERT and FastText Representations on Crowdfunding Campaign Success Prediction”
1. The abstract provides a clear overview of the study, detailing the relevance of crowdfunding, the application of AI, and the comparative analysis of BERT and FastText representations. However, it could be more concise. The mention of exploratory analysis and specific models (LSTM, GBM) is appropriate, but the abstract should avoid technical details and focus on the primary findings and significance. Simplifying the language to highlight the main contributions and results would enhance readability.

Experimental design

2. Simplify technical jargon and focus on key results.
3. Summarize the main findings and their implications in a straightforward manner.
4. The introduction successfully contextualizes crowdfunding within the post-2008 economic landscape and explains its importance as an alternative funding model. The literature review is comprehensive, covering various factors influencing crowdfunding success. However, the transition to the study's specific focus on sentiment analysis and AI models feels abrupt.
5. Ensure a smoother transition from general crowdfunding background to the specific research focus.
6. Explicitly state the research gap this study addresses and its contribution to the field early on.
7. The methodology is detailed, describing the use of BERT and FastText for feature extraction, followed by LSTM and GBM models for classification. This section is well-structured but could benefit from more clarity on the preprocessing steps and the rationale for choosing specific layers of BERT.
8. Provide a clearer description of the preprocessing steps.

Validity of the findings

9. Justify the choice of specific BERT layers and explain why LSTM and GBM were chosen.
10. The results section presents the findings clearly, showing that BERT representations outperformed FastText and that fine-tuning BERT improved performance. However, it could include more comparative statistics and visual aids (e.g., charts, graphs) to illustrate performance differences more effectively.

Additional comments

11. Visualization: Include charts or graphs to visually represent performance metrics.
12. Provide a more detailed comparative analysis of BERT and FastText across different metrics.
13. The discussion interprets the results in the context of existing literature, noting the superior performance of BERT. It also highlights the novelty of using different BERT layers and fine-tuning. However, it lacks a critical examination of potential limitations and the implications of findings beyond the study.

·

Basic reporting

Overall, the paper makes a significant contribution to the field of crowdfunding prediction using advanced AI models. With the suggested improvements, it could offer even greater insights and practical applications.

Experimental design

1. Discuss the limitations of the study, such as dataset size or potential biases.
2. Explore the broader implications of the findings for the crowdfunding industry and future research.
3. The conclusion reiterates the study’s findings and compares them with related works. It effectively summarizes the contributions but could be more concise and focus on the practical implications and future research directions.
4.Conclusions need improvements. Summarize key findings more succinctly.
5. Future Research: Provide specific and actionable directions for future research.

6.General Critique:
The paper provides a thorough comparative analysis of BERT and FastText in predicting crowdfunding success. It is well-structured and grounded in relevant literature. However, there are areas for improvement, particularly in terms of clarity, conciseness, and the critical evaluation of methods and results.

7. Consistency: Ensure consistent terminology and explanation depth across sections.
8. Technical Jargon: Minimize technical jargon in non-methodological sections to enhance accessibility.
9.References: Ensure all references are current and relevant, considering recent advances in AI and crowdfunding research.

Validity of the findings

10. Justify the choice of specific BERT layers and explain why LSTM and GBM were chosen.
11. The results section presents the findings clearly, showing that BERT representations outperformed FastText and that fine-tuning BERT improved performance. However, it could include more comparative statistics and visual aids (e.g., charts, graphs) to illustrate .

Additional comments

As above

Reviewer 3 ·

Basic reporting

1. The language is clear and readable.
2. The section on related works is insufficient regarding the use of BERT as a classification embedding tool. This needs to be described in more detail in the related works section.
3. The dataset is not introduced in sufficient detail. In the first chapter, many vectors that could influence success or failure were described, but only campaign definitions were used in this research.
4. The hypothesis that campaign definitions lead to successful prediction seems weak.

Experimental design

1. The details of LSTM and GBM are too lengthy.
2. The discussion on pre-trained BERT and FastText is insufficient. The paper should explain why these pre-trained models were chosen over others (many different pre-trained models can be found on GitHub/Huggingface). The specific pre-trained models used in the research are not detailed. The size of the training data used in the pre-trained models and the differences between BERT and FastText will significantly influence the results.
3. LSTM is a kind of RNN, so the sequence of the words (vectors) fed into LSTM matters. This paper selects 25 words from the campaign definitions, but it does not clarify the order of the 25 words in the 25*768 vectors. The method used in this paper is very different from past research papers. The authors need to address why they did not input all campaign definitions into BERT with LSTM.
4. SVM is mentioned but not explained in the main text. The purpose of mentioning SVM should be clarified.

Validity of the findings

1. Recall, Precision, and F-Measure are described in the manuscript, but recall is not used. Why?
2. The highest accuracy achieved in this research is 0.745, but there is no discussion on whether this is good or bad. No past methods are compared to the proposed method. The basic accuracy of a two-class classification is 0.5. If we always guess "success" in this case, the accuracy is around 0.6. The authors need to discuss the results.
3. Why calculate the mean of all layers of BERT or FastText? The authors need to discuss this choice.

Additional comments

No

---

## Round 0.2 · accepted · Accept

Dear authors,

Thank you for the revision and for clearly addressing all the reviewers' comments. I confirm that the paper is improved. Your paper is now acceptable for publication in light of this revision.

Best wishes,

·

Basic reporting

My Recommended changes are properly incorporated. I recommended this article for Publication

Experimental design

My Recommended changes are properly incorporated. I recommended this article for Publication

Validity of the findings

My Recommended changes are properly incorporated. I recommended this article for Publication

Additional comments

My Recommended changes are properly incorporated. I recommended this article for Publication

·

Basic reporting

My Recommended changes are properly incorporated. I recommended this article for Publication

Experimental design

My Recommended changes are properly incorporated. I recommended this article for Publication

Validity of the findings

My Recommended changes are properly incorporated. I recommended this article for Publication

Additional comments

My Recommended changes are properly incorporated. I recommended this article for Publication

·

Basic reporting

Manuscript ID Submission ID 88691v2
This paper is related to reviewing the manuscript (Version-2) titled " Comparative analysis of BERT and FastText representations on crowdfunding campaign success prediction"
Through a close examination of the language content of Kickstarter projects, this study aims to investigate how advanced artificial intelligence techniques may be used to forecast project outcomes. In order to do this, the effectiveness of two well-known text encoding frameworks—BERT and FastText—when combined with LSTM and GBM classification algorithms is compared.

Experimental design

This is the second version of the article. It is seen that the author has examined all the revisions claimed, requested and suggested by the editor and 3 referees in the first revision, item by item, and has made the necessary edits carefully and meticulously.
All requests have been made in the Basic Reporting, Experimental Design and Validity of the Findings sections raised by the editor.

Validity of the findings

All requests have been made in the Basic Reporting, Experimental Design and Validity of the Findings sections raised by the editor.
a) The author gave an explanation for the selection of LSTM and GBM and defend the use of particular BERT layers.
b) In the methods section, the author condensed the explanations of LSTM and GBM to highlight the key elements pertinent to this investigation.
c) To highlight performance discrepancies, comparative data and visual aids—such as bar charts for performance metrics—have been introduced to the results section.
d) The Conclusion and Discussion section provides a clear summary of the key results and their implications.
e) In order to create a more seamless transition from the overview of crowdfunding to the particular focus on sentiment analysis and AI models, the author included transitional phrases in the Introduction.
f) According to the Reviewer suggestion, Recall was incorporated into the assessment metrics by the author, and in the results section, a more thorough examination of precision, recall, and F-Measure was given.
g) Finally, the question asked by Referee 3, "The accuracy score given by the method was not interpreted" was answered and the necessary result evaluation was added to the paper.

Additional comments

For these reasons above;
My decision is accepting. I do not see any harm in publishing the manuscript.
Best regards.